

# Season-Ahead Forecasting of Water Storage and Irrigation
# Requirements
*An Application to the Southwest Monsoon in India*
**Arun Ravindranath[1]∗, Naresh Devineni[1], Upmanu Lall[2], Paulina Concha Larrauri[3]**
[1]Department of Civil Engineering , Center for Water Resources and Environmental Research (City Water Center),
NOAA Center for Earth System Sciences and Remote Sensing Technologies , City University of New York (City
College), New York, NY 10031
[2]Department of Earth and Environmental Engineering , Columbia Water Center, The Earth Institute, Columbia
University, New York, NY 10027, USA
[3]Columbia Water Center, The Earth Institute, Columbia University, New York, NY 10027, USA
**Short title**: A novel crop water stress index

19 ∗ - Contact Author's email: aravind000@citymail.cuny.edu



**Abstract**
Water risk management is perhaps the most ubiquitous challenge a stakeholder in the
water or agricultural sector faces. We present a methodological framework for forecasting water
storage requirements and present an application of this methodology to risk assessment in India.
The application focused on forecasting crop water stress for potatoes grown during the monsoon
season in the Satara district of Maharashtra. Pre-season large-scale climate predictors used to
forecast water stress were selected based on an exhaustive search method that evaluates for
highest Rank Probability Skill Score and lowest Mean Squared Error in a leave-one-out cross
validation mode. Adaptive forecasts were made over the years 2001 through 2013 using the
identified predictors and a semi-parametric k-nearest neighbors approach. The accuracy of the
adaptive forecasts (2001-2013) was judged based on directional concordance and contingency
metrics such as hit/miss rate and false alarms. Based on these criteria, our forecasts were correct
nine out of thirteen times, with two misses and two false alarms. The results of these drought
forecasts were compared with precipitation forecasts from the Indian Meteorological Department
(IMD). We assert that it is necessary to couple informative water stress/risk indices with an
effective forecasting methodology to maximize the utility of such indices, thereby optimizing
water management decisions.
**Keywords:** Crop stress, water risk, seasonal forecasts, climate-information, deficit, monsoon
prediction, contract farming, agricultural drought risk


















## 1. Introduction

Monitoring and forecasting systems can aid in pinpointing mitigation tactics for water security and water resources management. There is a continued interest in forecasting and monitoring systems that can inform planners and decision-makers in various water-dependent sectors at sufficient lead times and with increasingly higher levels of accuracy and reliability. The agricultural sector is perhaps the greatest example of this, being a heavily water-dependent sector that serves as the economic backbone of a country. The agricultural sector consumes more freshwater than any other economic sector, with an estimated 1,300 m$^3$/cap/yr needed to maintain an adequate diet (Rockstrom et al., 2009). Significant increases of water will be required to produce food by 2050, ranging from 8,500 to 11,000 km$^3$/yr, depending on to what extent rainfed and irrigated agricultural systems improve (Rockstrom et al., 2009). Additionally, to maintain high yields, irrigation will continue to be an important buffer to climate shocks. This is especially true when one considers that almost all of the world's major agricultural lands are located in the most drought-prone areas of the world (Mishra and Desai, 2006). Hence, developing forecasting techniques to improve how we address irrigation requirements, water storage requirements and crop water stress is a major step in dealing with the larger issue of water resources management at local, regional and global scales. The present study focuses on forecasting water storage and irrigation requirements in the agricultural sector as one important dimension to the larger issue of drought forecasting and water resources management, with an application of such forecasting to the monsoonal climate of India.

Existing forecasts either deal directly with basic hydrologic or meteorological variables, such as precipitation, temperature and soil moisture, or they work with proxies of drought, often in the form of indices such as the Standardized Precipitation Index, or SPI (McKee et al, 1993), the Palmer Drought Severity Index, or PDSI (Palmer, 1965), the Standardized Precipitation Evapotranspiration Index, or SPEI (Serrano et al, 2010), and the Normalized Difference Vegetation Index, or NDVI, among others. A comprehensive list of indices used in drought forecasting can be found in Heim (2002), Mishra and Singh (2010) and Liu and Pan (2016). The forecast of basic variables requires subsequently integrating these forecasts into a product that can estimate water storage or irrigation requirements, as these variables do not immediately divulge such information. This represents a challenge by itself. In light of this limitation, in this paper, we present a crop water stress index that is defined and constructed based on work by Devineni et al (2013). The advantage of this particular index, hereby known as the cumulative deficit index (CDI), is that it accounts for the variability in water supply and demand while incorporating information specific to a particular crop of interest. CDI is derived by accumulating differences in supply (rainfall) and demand (crop water requirement), and with very few crop input parameters. The CDI is a determinant of water stress faced by the crop and hence of the dependence of the crop yield on water availability. It can be interpreted as the water that is required from external storage beyond rainfall to meet demand (Devineni et al, 2013; Devineni et al, 2015). Therefore, the index directly informs water storage and irrigation requirements.

The primary focus of this paper will be on exploring the possibility of providing forecasts for CDI by investigating the sources of predictability and developing statistically verifiable models for the season-ahead probabilistic forecasts. Significant crop water deficits can adversely impact the crop production or water reserves and lead to high-energy costs for pumping



groundwater for irrigation to maintain yield. The seasonal forecasting of CDI provides a way for institutional planning and action in this context to reduce the climate-related water risks in agriculture, which is one of the largest consumers of water. An application of CDI forecasting is presented for the state of Maharashtra in India to verify whether advance reliable forecasts for potato-based CDI can be developed. A semi-parametric k-nearest neighbor (kNN) bootstrapping algorithm as described in Lall and Sharma (1996) is employed for forecasting CDI using pre-season large-scale climate indices. This is a simple probabilistic forecasting procedure that captures uncertainty. We examine these forecasts and suggest ways of interpreting them in a manner that can aid stakeholders in the agricultural water resources sector in addressing the fundamental questions about irrigation and water storage requirements. These forecasts will then be compared to precipitation forecasts for the same season in the same area of India as given by the Indian Meteorological Department (IMD).

In section 2, we present a survey of the existing forecasting systems in monsoonal climates and their skill and limitations. In section 3, we discuss the background and scientific basis of CDI, including its explicit formulation and governing equations. In section 4, we get into a thorough description of the case study and all steps involved, including background information relating to the case study and location, data collection and processing, a complete description of the forecasting model, methods and predictor selection scheme. Section 5 presents the results of the forecast, a discussion of these results and their implications, and a comparison of our results with those of IMD. Finally, section 6 summarizes and concludes the paper.

## 2. A Brief Review of the Current Forecasting Systems for Water Management in Monsoonal Climates

A number of forecasting methodologies have been proposed or developed for water management and agricultural planning. Shah and Mishra (2016) investigated the goodness of the Global Ensemble Forecast System (GEFS) for generating medium-range (~7 day) drought forecasts in India, and found that the GEFS has higher forecasting skill during the non-monsoon season than monsoon season for both temperature and precipitation, largely due to intraseasonal variability during the monsoon season. This forecasting system tends to forecast temperature variables with higher skill than precipitation and has variable skill according to region. Hence, there is sensitivity to intraseasonal variation, which monsoon climates are notorious for, and regional variation as well. Mishra and Desai (2005) used well-chosen linear stochastic models (ARIMA) to forecast SPI- 3, 6, 9, 12, and 24 as a drought proxy in the Kansabati River Basin, an important source of water for irrigation and an area in which crops are grown, in the Purulia district of West Bengal, India at lead times of 1, 2, 3, 4, 5 and 6 months. Highest skill, as measured by the correlation coefficient between observed and model-predicted SPI series, occurred at shorter lead times, with correlation values between 0.799 and 0.925 depending on which SPI series was forecasted. Asoka and Mishra (2015) forecasted vegetation anomalies (as NDVI) at the regional scale as a proxy of vegetation health, and thus moisture availability. The model used NDVI, root-zone soil moisture, and sea surface temperature (SST) at one to three months lead time to develop the vegetation anomaly forecast, and skill was highest at one month lead time and much lower for two and three months lead time as measured in a validation phase by examining the $R^2$ statistic and by plotting the observed NDVI against the model-interpolated series for one-, two-, and three- month lead times. Skill also varied based on location in space and, in particular, was lower during the monsoon season (JJAS) likely due to the effect of





intraseasonal variability of the monsoon system on agricultural practices. Belayneh and Adamowski (2012), in the interest of drought forecasting, forecasted SPI 3 and SPI 12 over lead times of one and six months in the Awash River Basin in Ethiopia using Artificial Neural Network, Wavelet Neural Network and Support Vector Regression models and similarly found that forecast skill was higher at the shorter lead time. Kar et al (2012) considered Multi-Model Ensemble (MME) methods in both a deterministic and probabilistic context. It was found that the individual member models showed poor skill in simulating monsoon interannual variability and that on average spatially, a MME scheme that uses the member models as predictors in a point-by-point multiple regression as a means of averaging the member model forecasts outperforms the other schemes mentioned in the paper in forecasting precipitation. However, it was found that even here, none of the three MME schemes had any usable skill in a certain region of India, and it was concluded that a probabilistic system would work better. When probabilistic forecasts were generated (probabilistic MME) and evaluated for skill, RPSS was positive for the best scheme, in only the northern most parts of India and a few scattered points in north and central India. Finally, Shah et al (2017) examined how different forecast products can be used operationally to provide hydrologic forecasts (e.g. for precipitation, temperature) for India at a 7 – 45 day accumulation period, which is critical for agricultural and water resource planning. Forecast skill was evaluated on the basis of correlation with observations, median absolute error (MAE) and the critical success index (CSI). Four forecast products from Indian Institute of Tropical Meteorology (IITM) were compared with Climate Forecast System version 2 (CFSv2) and Global Ensemble Forecast System version 2 (GEFSv2) forecast products, and it was found that the meteorological variables predicted from the IITM products showed superior skill for all accumulation periods. The key point here is that the IITM ensemble is postulated to capture intraseasonal variability of rainfall during the monsoon season.

As an alternative to these agricultural planning measures, we introduce a new seasonal crop water stress index that is more informative than the total rainfall measure. It gives a surrogate for irrigation water required and incorporates intraseasonal rainfall and temperature variability along with information inherent to the specific crop and planting region.

## 3. The Cumulative Deficit Index: Background and Scientific Basis

Our interest in this study is to provide one-season-ahead forecasts of irrigation and water storage requirements for water resources management in the agricultural sector, and subsequently compare the outcomes of these forecasts with the forecasts issued by IMD. We begin by developing an index for crop water stress as a means of gauging irrigation requirements. The index developed and used in this study computes the maximum cumulative deficit over a growing season between daily water requirement for optimal crop growth and daily effective rainfall. Variants of this method have been presented in our previous studies for quantifying the water stress globally (Devineni et al, 2013; Devineni et al, 2015; Chen et al, 2014), and drought indexing for the United States (Etienne et al, 2016; Ho et al, 2016). Given an *n*-year record of daily data, our water stress index calculates the day-by-day accumulation of deficit in rainfall in each of the *n* growing seasons. The maximum of these seasonal daily deficit values is taken to be the value of the index for the season. Hence, we give this index the name *cumulative deficit index*, abbreviated CDI. On a practical level, such an index gives a worst-case scenario in terms of the seasonal water stress on the crop, and can therefore be interpreted as the amount of water that should be drawn from external storage to meet water demand. This





may include irrigation, ground water pumping, interbasin transfers, and/or withdrawing water
from a storage or water-harvesting facility.
Deficit is estimated as the difference between the seasonal crop water requirement and
effective rainfall for each crop in a given location in the season. Effective rainfall is given as

$$S_{t,d} = \alpha * P_{t,d} \ \dots \ (1)$$

In Eq. (1), $P_{t,d}$ is the rainfall for a given day $d$ in the year $t$. $\alpha$ is the parameter that determines
the fraction of rainfall that can be utilized by the crops for a location. It accounts for losses to
direct runoff, evaporation and groundwater infiltration. In our study, we set $\alpha = 0.7$.
The water use for a given crop is estimated based on the expected growth stage and daily
evapotranspiration as

$$D_{t,d} = k_{c,d} * ET_{0 \ t,d} \ \dots \ (2)$$


In Eq. (2), $k_{c,d}$ is the crop coefficient, which is the ratio of actual evapotranspiration ($ET_a$) of a
given crop under non-stressed conditions to reference crop evaporation ($ET_0$). It represents crop-
specific water use at various growth stages of the crop and is typically derived empirically based
on local climatic conditions (Doorenbose and Pruitt, 1977). The accumulated deficit over a
season is then given as

$$deficit_{j,d} = \max(deficit_{j,d-1} + D_{j,d} - S_{j,d}, 0) \ \text{where} \ deficit_{j,d=0} = 0 \ \dots \ (3)$$


$$CDI_{j,t} = \max\big(deficit_{j,d(y)}: d = 1: n_s; t = 1: n\big); \text{where} \ deficit_{j,d(0)} = 0, \ y=1,\dots,n \ \dots \ (4)$$


In equation (3), $deficit_{j,d}$ refers to the accumulated daily deficit for any given year with a crop
growth period of $n_s$ days in the year, $D_{j,d}$ to total daily water demand, $S_{j,d}$ to the total daily
effective rainfall, for geographical location $j$, and day $d$; $t$ refers to a calendar or cropping year;
and $n$ is the total number of years in the analysis. For an $n$-year record, seasonal water stress is
evaluated as the maximum cumulative deficit each season and defined here as $CDI_{j,t}$. CDI
focuses on the rainfall distribution within the season relative to the crop water demand. It
therefore accounts for the timing of planting, different stages of crop growth, and the timing and
distribution of rainfall in the season. The index may also be treated as a hydrologic index and
forecasted exactly as one would forecast precipitation or temperature variables, or any other
water stress or drought index. Depending on the lead time of such forecasts, this can give
farmers and other agricultural stakeholders a sufficient amount of planning and preparation time,
thus providing them a critical edge in hedging agricultural water risk. This is critical in irrigation
and water storage planning.

**4. Case Study: Forecasting Irrigation Requirements for Potatoes in Maharashtra, India**

We endeavored to forecast CDI for potatoes grown in the Satara district in Maharashtra,
India as an application. The Satara district in Maharashtra is one of the primary regions for
sourcing potatoes during the monsoon season (June - September). Satara supplies the majority of
the potatoes processed by the Frito-Lay manufacturing plant in Pune, Maharashtra (Economic
Times, 2013). Potato is a major cash crop in Maharashtra and accounts for at least 75% of total





production (Nikam, *et al*., 2008). The average annual rainfall in this arid to semi-arid region is around 350 mm with high inter-annual variability. The region has experienced four droughts (seasonal rainfall below long-term average) since 2001. The ability to predict such droughts with a reasonable accuracy at lead times of three to six months could suggest ways to adapt existing agricultural operations to the anticipated conditions and minimize the impacts of droughts on the agricultural supply chain. Hence, we develop, present and evaluate the results from retrospective forecasts of CDI for the monsoon season over the period 2001-2013.  The June-July-August-September (JJAS) season is the growing season for potatoes in the Satara district. It is also the core monsoon season for the Indian sub-continent.  The forecasts use climate data from three to six months prior to the beginning of the monsoon season as predictors, and forecasts are to be issued in May, one month prior to monsoon onset.

### *4.1: Data Collection and Processing*
#### *4.1.1: Precipitation and Temperature Data and the CDI*
Gridded daily rainfall data from 1901 – 2004 available at $1^0$ x $1^0$ spatial resolution from the India Meteorological Department (Rajeevan *et al*., 2006), and gridded daily temperature data from 1948 – 2000, available at the same spatial resolution from National Center for Atmospheric Research (Ngo, *et al*., 2005) are used in this study. Since the daily temperature data is available only for 53 years, we used the daily climatology, i.e. the mean daily temperature, for the remaining 60 years (Devineni *et al*., 2013). The daily climate time series grids were spatially averaged over the Satara district. This process resulted in a time series of daily precipitation and temperature estimates for 104 years. The daily Reference Crop Evapotranspiration ($ET_0$) was developed based on the daily time series of minimum, mean and maximum temperature data, and extraterrestrial solar radiation (Hargreaves and Samani, 1982). The Hargreaves method is used globally to predict $ET_0$ in regions where data availability is limited to air temperature data (Allen, *et al*., 1998). Seasonal daily rainfall data from 2005 to 2013 for the Satara district were collected separately from a website maintained by the Agricultural Department of Maharashtra State and used to augment the 104 years of rainfall and temperature data. The CDI was computed for each of these 113 seasons using the daily rainfall data and reference crop evapotranspiration.  This will serve as the predictand for our forecast model.  The computation of CDI is illustrated in Fig. 1.  These figures provide insights on the time-evolving vulnerability to stress arising from deficient rainfall and changes in crop demand.

CDI as a water stress measure is a proxy of not only crop water stress but also irrigation and water storage requirements.  Consider Fig. 1.  When daily seasonal rainfall is low or when rainfall enters an inactive phase for a considerable period of time, as displayed by the vertical cyan bars, the amount of daily accumulated water deficit increases to reflect the disparity between water supplied as rainfall and the water required by the crop to sustain itself, as displayed by the red curve in Fig. 1.  The highest point, or peak, on the black deficit time series in Fig.1 is the value of CDI, and it prepares us for the worst-case scenario of deficient water supply for the crop.  This can be calculated for multiple crops, each CDI value depending on the specific crop's water demand and the location and time of planting.  This gives the stakeholder a conservative estimate of how much additional water is needed beyond what Nature is willing to supply in order to maintain critical yields while apportioning water resources intelligently.  Since agriculture tends to be one of the largest consumers of water --- about seventy-percent of all the





world's freshwater withdrawals go towards irrigation use (USGS, 2017), and this is in addition to
what is rainfed --- this is an integral part of water resources management.
The annual time series of the CDI computed for the JJAS season (referred to as Kharif
season in India sub-continent) in Satara is presented in Fig. 2. We have standardized the CDI
values as the percentage difference each year from the 113-year average of CDI. The long-term
average CDI for growing potatoes in Satara is 241 mm. This is equivalent to approximately
257,644 gallons of water used for irrigating a one-acre farm of potatoes on average throughout
the season. The percent differences in Fig. 2 refer to percentages of this number, i.e. a 10%
increase in CDI indicates an additional requirement of 25,764 gallons. From Fig. 2, it is clear that
(a) Satara experiences recurrent droughts with intermediate wet periods and (b) there is year-to-
year persistence in the incidence of these droughts. Such variations and epochal changes are
typically modulated through large-scale global climate patterns. Investigating the relationship
between monsoon deficit and the large-scale climate teleconnections could enable the
development of models that can be used to understand and predict the variability in the CDI in
the region.
*4.1.2: Climate Precursors and Climate Data*
Our goal was to develop a simple statistical model for predicting CDI for potatoes grown in
Satara. The generalized climate forecast models available at low spatial resolution are not
specific enough for this task. Consequently, the first objective was to identify appropriate climate
predictors before the monsoon starts in June. There is an extensive history of developing long-
range predictions of monsoon rainfall that are based on various regional to large-scale climate
predictors (Walker, 1924; Thapliyal, 1987). A variety of seasonal forecasts of the all India
Summer Monsoon Rainfall (ISMR) are documented and available for reference (Gadgil et al.,
2007; Kumar et al., 1995).
It is well established that inter-annual climate modes such as ENSO associated with
anomalous Sea Surface Temperature (SST) conditions in the tropical Pacific Ocean influence the
inter-annual variability of ISMR (Parthasarathy and Pant, 1985; Shukla and Paolino, 1983).
Anomalously warm tropical eastern Pacific SSTs (El Niño) are associated with a drier-than-
normal ISMR, whereas anomalously cool tropical eastern Pacific SSTs (La Niña) are associated
with a wetter-than-normal ISMR (Sikka, 1980; Parthasarathy and Panth, 1985; Rasmusson and
Carpenter, 1983). Ihara, *et al*. (2007) have suggested that the ENSO warm (cool) phases shift the
location of the tropical Walker circulation and cause deficient (excessive) rainfall by suppressing
(enhancing) the convection over India. Hence, ENSO indices were chosen to be among the
candidate predictors for the forecast model.
Raw monthly SST data for the Niño 3, Niño 4, Niño 12 and Niño 34 indices were taken from
the KNMI climate explorer database (KNMI, 2016). For each given raw ENSO index (3, 4, 12
and 34), we considered three different types of derived ENSO indices: a December-January-
February (DJF) seasonal average, a March-April-May (MAM) seasonal average, and a MAM
minus DJF (MAM-DJF) differenced time series. Among the Niño indices calculated, the change
in the tropical Pacific SSTs from December to May (MAM-DJF trend) was found to be of
significance by previous investigators.  Shukla and Paolino (1983) found the correlation
coefficient between the MAM-DJF trend pressure anomalies and the ISMR to be a significant -
0.42.  Parthasarathy et al. (1988) found the correlation coefficient between this winter-to-spring
trend and ISMR over the period 1951-1980 to be between 0.40 and 0.52 in magnitude, depending
on the specific region within the tropical pacific.  Hence, MAM-DJF trends from Niño 3, Niño 4,



Niño 12 and Niño 34 were considered to be potential model predictors. Parthasarathy et al.
(1988) found that the MAM-averaged tropical Pacific SSTs over the box 14 N to 20 N, 176 E to
160 W had a correlation of -0.40 with ISMR, convincing us to consider this average as well. In
addition to the MAM and MAM-DJF averages, we computed the winter season (DJF) average,
although DJF-averaged tropical Pacific SSTs were not found to be significant in the literature.
However, it is worth noting that Parthasarathy et al. (1988) found that the correlation coefficient
between the Darwin SLP during the DJF season and ISMR was +0.39.
As the concurrent season (JJAS) state of ENSO has an important, well-documented impact
on ISMR, we also elected to include the Niño 34 JJAS average. As mentioned earlier, an El
Niño event during the JJAS season is strongly associated with an anomalously dry JJAS rainfall
season in India, while a La Niña event during the JJAS season is strongly associated with an
anomalously wet JJAS rainfall season in India, prompting our choice. We coupled the JJAS
seasonal average for the Niño 34 index with forecasts of the JJA and JAS seasonal averages for
the Niño 34 index. These forecasts were obtained from the International Research Institute for
Climate and Society (IRI) ENSO forecast page and covered the period 2002-2013. These
forecasts can be used to forecast JJAS monsoon CDI in place of the observed Niño 34 JJAS
values on a real-time basis. These forecasted values were averages of the projections from at
least six distinct statistical/dynamical models, with one average for the JJA season and one
average for the JAS season. Together, we start with a total of thirteen ENSO-based indices.
Other candidate predictor variables include concurrent season (JJAS) eastern Indian Ocean
SSTs known as the Indonesian Throughflow, or ITF. Warm, low-salinity water from the Pacific
is introduced into the Indian Ocean via the ITF and is considered to be an integral component in
the heat and hydrological budget of the Indian Ocean (Gordon et al., 1997). The ITF waters are
also believed to influence SSTs and associated ocean-atmosphere coupling within the Indian
Ocean, making it an important aspect of monsoon climate research (Gordon et al., 1997). Thus,
the ITF was also selected to be a candidate predictor in the model. During the JJAS monsoon
season, the ITF is strengthened considerably, allowing an abundant amount of relatively warm
water to be injected into the Indian Ocean. Eastern Indian Ocean SSTs during the JJAS season
correspond to enhanced (suppressed) atmospheric convection during the anomalous warming
(cooling) of the Indian Ocean waters, which in turn supplies (robs) the developing monsoon of
much-needed moisture. We found that the Spearman rank correlation coefficient between CDI in
Satara and the average SST anomalies over $20^o$ N and $5^o$ S and $100^o$ E and $130^o$ E (the region
representing ITF) during the JJAS season is around -0.35 (statistically significant at the 95%
level), suggesting that warm conditions in the ITF region result in below-normal CDI, or low
crop water stress. Figure 3 presents the field correlation map of SST anomalies with CDI. For
these reasons, we chose concurrent season ITF data to be a candidate predictor. The ITF data was
collected from the IRI data library and consists of two components: an observation component
and a forecasted component. The observations consist of measured eastern Indian Ocean SST
anomalies during the JJAS season from 1901 through 2013. The forecasts consist of JJAS-
season ITF values retrospective from the ECHAM4.5 global climate model and cover the period
2001-2013. Skillful forecasts for the tropical SSTs based on coupled ocean-atmospheric general
circulation models have been in operation from various climate centers since 1998. Hence, in
the forecasting scheme, we used the ITF derived from forecasted SST state issued in May from
ECHAM4.5 operational forecasting center (available from IRI data library:
http://iridl.ldeo.columbia.edu/SOURCES/.IRI/.FD/.ECHAM4p5/.Forecast/.ca_sst/.ensemble24/;
Li and Goddard, 2005; van den Dool, 2007; Roeckner et al., 1996). The observed JJAS ITF data





are used to train the model, while the retrospective JJAS ITF forecasts are used to make forecasts
for the years 2001 – 2013.
### 4.2: The Forecasting Procedure
*4.2.1: Predictor Selection*
Given a pool of candidate predictors, the next step is to select the best subset of those
predictors. The predictors used in the forecasting model were chosen based on an exhaustive
search method.  In the exhaustive search method, all possible combinations of the candidate
predictor variables are used to develop models that are cross-validated on historical data. Skill
metrics are then used to compare the predictive accuracy of each combination. In the present
study, we began with 113 years of CDI data and fourteen candidates: Niño 3 DJF, Niño 3 MAM,
Niño 3 MAM-DJF, Niño 4 DJF, Niño 4 MAM, Niño 4 MAM-DJF, Niño 12 DJF, Niño 12
MAM, Niño 12 MAM-DJF, Niño 34 DJF, Niño 34 MAM, Niño 34 MAM-DJF, Niño 34 JJAS
and ITF.  The exhaustive search method utilized the kNN cross-validation algorithm and forty
years of training data (1901-1940) to build forecast distributions for each of the years 1941-2013.
At each step, the training data was updated to include data from all of the years up until the year
being cross-validated. Thus, we always use only the historical data and update the model each
year with the information of the previous year, much as a regular user of the forecast system
would have to do. These forecasting distributions, built over a 73-year record (1941 to 2013)
were created successively for every unique combination of two variables, every unique
combination of three variables, so on and so forth until we reached the entire pool of predictors.
For each and every possible unique combination of the predictor variables, we obtain a
matrix of seventy-three columns.  For each of these seventy-three (73) years, the squared error
and rank probability score (Epstein, 1969; Murphy, 1969, 1971; Candille and Talagrand, 2005)
were computed, and from this the root mean squared error (RMSE) and rank probability skill
score (RPSS) were computed.  In this manner, a single RPSS value and MSE value were
calculated for every possible combination of the predictor variables.  We chose the following
combination of predictors based on the relative optimality of both their RPSS and RMSE scores:
Niño 12 MAM-DJF, Niño 34 MAM-DJF, and ITF, and this set of variables had an RMSE of
49.25 mm of required (JJAS) seasonal water storage and RPSS of 0.26.  We devised a simple but
effective decision rule for determining the optimal choice of predictors based on ranking the
metric values.  This is especially useful when the number of combinations of variables is
unwieldy.  Optimality was determined by assigning a rank number to the RMSE and RPSS
values in such a way that the number 1 was assigned to the lowest RMSE value, 2 to the second
lowest RMSE value, and so on, and the number 1 was assigned to the largest RPSS value, 2 to
the second largest RPSS value, and so on.  For a fixed number of cross-validated predictor
candidates, and for each RMSE/RPSS pair, one pair for each combination of predictors, we
determined an RMSE and RPSS rank and took the sum of these ranks.  The smallest of all of
these sums corresponds to the best or optimal set of predictors among all possible sets of cross-
validated predictors.  We then compared the rank sum along with the number of predictors to
choose the best set of predictors.  The chosen trio of predictors mentioned above had the
unequivocally highest value of RPSS and second lowest RMSE value out of all possible
combinations of the original set of seventeen candidates, the lowest RMSE being only slightly
smaller at 48.92 mm.  Conceptually, this procedure is similar to the "best subsets regression" or
"step-wise regression" (Helsel and Hirsch, 2002), but in the spirit of using kNN algorithm for
forecasting, we designed this selection scheme to use the kNN algorithm instead.



CDI forecasts were subsequently made using the selected set of predictors.  The forecast
procedure is tested using the leave-one-out cross-validation method.  Each historical observation
is omitted in turn, and the model is developed using the remaining years of data.  A prediction of
the observation that was not kept in the model-building set is then made and compared with the
actual outcome for that year.  Results from a variant of this approach are presented in the next
section.  The CDI for the 2001 Kharif season is predicted using the model developed based on
data from 1901 – 2000.  Similarly, the CDI for 2002 is predicted based on the model that is
developed using the data from 1901 – 2001.  Thus, as we move from year to year, we update the
model observations and predict the future state.
*4.2.2: The k-Nearest Neighbors Real-Time Forecasting Model*
The forecasts were developed using a semi-parametric *k*-nearest neighbors (k-NN)
model.  This is a data-driven approach that develops a conditional probability distribution of the
CDI given the predictors by first identifying the *k*-historical climate conditions that are most
similar to the current values of the climate predictors and then randomly drawing the vector of
CDI values in the historical data that correspond to these *k* neighbors.  The neighbors are
weighted so that the closer or more similar neighbors are chosen more often than those further
away.  The key steps are as follows.
Let $\mathbf{X}$ be the design matrix of size *n* x *p*, where *p* = number of predictors selected from
the original pool of candidates.  Let $\mathbf{x}_i$ denote the $i^{th}$ row of $\mathbf{X}$.  Hence, $\mathbf{x}_i$ is a vector containing
the values of each of the *p* predictor variables during year *i*.  Denoting the current values of the
predictors by $\mathbf{x}_c$, the idea is to find *k* such predictor vectors from the historical record (i.e. find k
values of $\mathbf{x}_i$ with *i < c*) that are most "similar" to the value of $\mathbf{x}_c$ and use this information to
construct a sampling distribution of CDI from which we can issue probabilistic forecasts. The
number of neighbors in the model, or *k*, represents the number of degrees of freedom in the
model, and should be chosen with care, as the choice of *k* affects the skewness and level of
uncertainty in the sampling distributions.  After trying several different values for *k*, we found an
optimal value to be *k* = 25.  Rajagopalan and Lall (1999) recommend that *k* be roughly equal to
$\sqrt{n}$, where *n* = the total number of observations.  In our situation, it was evident that we required
more neighbors than this rule would allow, due to the skewness and variance apparent in the
sampling distributions when using only eleven or fewer neighbors.
Let $\mathbf{y}$ be the n-dimensional vector of seasonal CDI values, each component of which
represents the aggregate water deficit level over the JJAS growing season of every year in the
historical record.  Assume that $\mathbf{y}$ has been centered and normalized by its historical average to
produce mean-normalized anomalies.  The first step was to consider the individual distance
values (under some specified metric) between $\mathbf{x}_c$ and $\mathbf{x}_i$ for *i* = 1,...,*c*-1.  The chosen distance
metric for our k-NN model was the Mahalanobis distance (Mahalanobis, 1936)

$$D_M(\boldsymbol{x}_c,\ \boldsymbol{x}_i) = \sqrt{(\boldsymbol{x}_{c-}\boldsymbol{x}_i)^T \Sigma^{-1}(\boldsymbol{x}_{c-}\boldsymbol{x}_i)} \ \dots (5)$$

where $\Sigma$ is the covariance matrix of the training values in $\mathbf{X}$. The Mahalanobis distance measure
judges point separations in a metric space based on statistical dissimilarity, as opposed to solely
physical distance.  Hence, the level of similarity between predictor values across different years
is determined by the orientation and location of each point relative to the scatterplot of the
predictor data.  Large distances from $\mathbf{x}_c$ represent predictor values that are statistically anomalous
in the context of the predictor data.





After the Mahalanobis distances had been calculated, the $k$ (with $k = 25$) smallest distance
values were selected and the corresponding years in which these distances occurred were noted.
These years, hereby referred to as the *analog years*, are the years during which the predictor
signals were most similar to those of the current year. The vector-valued predictors during these
analog years are referred to as the *neighbors* of $\mathbf{x}_c$.
The final step was to resample CDI values from the analog years. The resampling
technique employed is a nonparametric method known as the *bootstrap* (Efron, 1979; Efron and
Tibshirani, 1993). The idea behind the bootstrap component is to sample with replacement from
a pool of data using the underlying distribution that generated the data to guide the sampling
process. We chose not to assign a parametric family of distributions to the CDI data, and instead
estimated its underlying distribution semi-parametrically using a kernel density estimator. This
semi-parametric method of k-NN bootstrapping was first introduced in Lall and Sharma (1996).
Applications of the methods using different variants have since been presented (see for example,
Rajagopalan and Lall, 1999, Souza and Lall, 2003 and references therein). We employed the
same discrete resampling kernel proposed in Lall and Sharma (1996), which has the general form
$K(j) = 1/(j*S)$ with $S = \sum_{j=1}^{k} 1/j$ , where j is the rank of each neighbor of $\mathbf{x}_c$, a rank of j=1
assigned to the closest neighbor and a rank of j=k assigned to the most distant neighbor. Our
strategy was to build this kernel density estimator based on the ranks of the selected neighbors
and resample the predictand values from these analog years. We resampled from the twenty-five
analog CDI values 1,000 times, and each of the twenty-five values was resampled proportionally
to the probability of its occurrence as determined by the density estimator.
*4.2.3: Analyzing the k-NN Results*
The way in which model results are interpreted and presented is important for potential
stakeholders. In this case study, our interest was in forecasting the CDI for a given potato
growing season in Satara. The information from these forecasts can be of great use to potato
farmers in Satara as well as corporations with investments in these farming areas. This
necessitates a clear and concise communication of the forecast results.
The output of the k-NN model was a time series for each forecasted year consisting of
1,000 realizations. This is the sampling distribution for the CDI and consists of mean-
normalized anomaly values from the analog years converted to percentage values. As stated in
the previous section, the deficit value from each analog year in the sampling distribution is
represented proportionally to its probability of occurrence as assigned by a kernel density
estimator. The sampling distribution is used to issue one-season-ahead probabilistic forecasts
(i.e. the likelihood of a deficit for the forthcoming growing season). There are a whole slew of
possibilities when it comes to using these sampling distributions for probability-based forecasts.
Our approach includes the following for a given forecasted growing season:
1. A boxplot depicting the sampling distribution with the observed percent anomaly value
superimposed on the boxplot for every growing season forecasted. In using predictand
anomalies, the historical mean becomes the zero line in the coordinate plane of the
boxplot.
2. A three-category forecasting system with the categories "above normal", "normal" and
"below normal", provided that the historical mean/climatology is the threshold that is
desired.





3. Calculate the probabilities for the categories specified in step 2 from the sampling distribution generated in step 1, and use this to evaluate the accuracy and strength of the forecast based on contingency metrics such as hit rates and false alarms.
4. To get a sense of the spread/variability in the boxplot distribution, calculate the Interquartile Range (IQR).
5. Compare the value of the observed percent anomaly of the predictand with the category in which the majority of the probability mass of the sampling distribution lies. This is of central importance in getting a basic sense of the accuracy of the forecast.

In general, the construction of such a sampling distribution allows the investigator the freedom to calculate probabilities on many different thresholds. The thresholds should be defined by the particular application and the needs of any stakeholders involved.

## 5. Case Study: Forecast Results and Discussion
### 5.1: CDI Forecast Results and Comparison with IMD Monsoon Forecasts

We hereby present the results of the CDI forecasts for the 2001 – 2013 JJAS seasons in the Satara district, Maharashtra, India. Forecasts are specifically made in the interest of irrigation requirements for potatoes grown in the Satara district, and we discuss the results in this context. The output of the k-NN model is the forecasting distributions for CDI of the thirteen years and a series of boxplots representing these forecast distributions as shown in Fig. 4. The probabilities calculated from these distributions are shown in Table 1, columns 2 and 3.

Figure 4 shows a series of boxplot diagrams depicting the k-NN forecast distributions for CDI over the years 2001 – 2013. All calculations in this Figure, including the construction of the distributions themselves, were done using anomalies of the predictand rather than the raw predictand values. The anomalies were calculated by subtracting the 1901 – 2013 mean from the data and dividing by this mean value and converting the quotient to a percentage. The idea is to gauge the level of seasonal crop water deficit in a forecasted year with respect to the level of crop water deficit that has occurred on average over the entire historical record. This should address the question: how "normal" or "abnormal" is a given level of deficit over a season with respect to everything we have seen or experienced thus far. Given that the forecast is developed one season ahead, the sign of a strong shift in the probability will alert the decision-makers to an anticipated deficit or surplus event.

We have created two general possibilities: the observed percent anomaly values (triangles in Fig. 4) can be positive or negative. As the forecasts have been carried out using anomalies instead of raw values, the 1901 – 2013 historical average is re-positioned as the zero line in Fig. 4. We calculate the probability under the kNN forecast distribution of observing positive (negative) deficit anomalies for each year in 2001 – 2013. These are retrospective forecasts in the sense that these anomalies have already been observed and recorded but not used in building the model. These probabilities, corresponding observed percent anomalies and IQR values are presented in Table 1. The utility of these forecasts are discussed in section 5.2.

Given the above information, we judge the accuracy of the forecasts during any given year on a few simple criteria: the directional agreement between the observed percent predictand anomaly and the median of the forecast distribution (Fig. 4), joint consideration of the forecast probabilities and the observed percent anomaly (Table 1, columns 2, 3 and 4) and the level of uncertainty in the forecast distribution (Fig. 4 and Table 1, column 5). Uncertainty is measured by the IQR of the boxplot distribution. In the present context, we say that a forecast for a given year has *identical directionality* (with respect to the observation) if both the median of this



forecast and the observation (as a percent anomaly) are either positive (above the historical
average) or negative (at or below the historical average).  The absence of identical directionality
will be called *dissimilar directionality*.
The box-and-whiskers plots shown in Fig. 4 for each year illustrates the range of possible
values of the CDI for that year.  We have identical directionalities for the years 2001, 2004,
2005, 2006, 2007, 2010, 2011, 2012 and 2013.  For the years 2001, 2011 and 2012, the model
correctly forecasted that the water stress conditions for the Maharastran potatoes would be above
the CDI climatology.  We can see from Fig. 4 that both the observed percent anomalies
(triangles) and the medians for all of these forecasted years are positive.  Additionally, Table 1,
column 2 shows that the majority of the probability mass of the kNN distribution is placed in the
"Above Mean" category for 2001, 2011 and 2012, while column 4 shows that for these years, the
observed CDI anomalies are positive.  Similarly, for the years 2004, 2005, 2006, 2007, 2010 and
2013, the model correctly forecasted that water stress conditions for the potatoes would be below
the historical average, and this can be seen from Fig. 4, where the observed anomalies and the
medians for all of these forecasted years are negative.  Similarly, Table 1, column 3 shows that
the majority of the probability mass from the kNN forecasting model was placed on the "Below
Mean" category for these years, and the corresponding observed CDI anomalies are also
negative.  For the years 2002, 2003, 2008 and 2009, we have dissimilar directionalities.  The
forecasts suggest higher probability values for below average CDI during 2002 and 2003,
whereas positive anomalies were observed for these years.  Similarly, the forecasts for 2008 and
2009 placed the majority of the probability mass on higher than average CDI, suggesting that
these years were likely to see higher than normal potato water stress.  However, the observed
CDI anomalies were negative, implying the opposite scenario.
We say that a *hit* has occurred if identical directionality is observed.  A *miss* occurs if the
forecast implies below average water stress, but the observation shows above average water
stress.  Finally, a *false alarm* occurs if the forecast implies above average water stress while the
observation shows below average water stress.  Table 2 shows that the hit rate of the kNN
forecasts is 9/13, the miss rate is 2/13 and the false alarm rate is 2/13.  Table 3 shows a
comparison of our CDI forecasts with seasonal total precipitation forecasts of the India
Meteorological Department, abbreviated IMD.  The IMD forecast presented here for 2001 is
long-range for precipitation in the JJAS season over three climatically homogeneous regions in
India: Northwest India, Peninsular India, and Northeast India.  Maharashtra is in Peninsular
India, and so we refer to this forecast.  For 2001, the forecast result was categorized as either
normal, above normal or below normal.  "Normal" is defined as being within ±10% of the long-
period average, or LPA.  Beginning in 2003, IMD began offering two-stage forecasts, the first
released in mid-April using data up to March and an update in June using data up through May.
For both 2011 and 2013, we used the initial country-wide forecast, as the updated forecasts for
JJAS could not be found.  In 2003, IMD began to divide their forecast results into five
categories: drought/deficient, below normal, near normal/normal, above normal and excess.
"Deficient" (drought) is defined as JJAS total seasonal rainfall that is less than 90% of the long
period average (LPA).  "Below normal" is defined as JJAS rainfall that is 90% – 96% of the
LPA, "normal" (sometimes called "near normal") is defined as JJAS rainfall that is 96% – 104%
of the LPA, "above normal" is defined as JJAS rainfall that is 104% – 110% of the LPA and
"excess" is defined as JJAS rainfall that is more than 110% of the LPA.  The IMD forecasts are
reported as percentages of the LPA, as shown in column 3 of Table 3.  Going by the categories
defined by IMD, and comparing these forecasts with actual JJAS seasonal total precipitation





anomalies from our gridded rainfall data set, where these anomalies have been calculated with
respect to the long period average defined as 1901 – 2013, we classify each forecast as a hit, miss
or false alarm as was done with the CDI forecasts. The hit rate for IMD is 1/9, the miss rate is
3/9 and the false alarm rate is 5/9. We must bear in mind that the total precipitation forecasts
given here are for an entire region that includes the state of Maharashtra, whereas our CDI
forecasts are generated based on CDI calculations from the target location of Satara,
Maharashtra, India. Hence, our CDI anomalies reflect the conditions of Satara on a much higher
resolution than the coarse IMD precipitation anomalies. Furthermore, we are comparing IMD
forecasts with actual precipitation totals from Satara, and computed with respect to the 1901 –
2013 LPA instead of the 1951 – 2000 LPA of IMD, under the reasonable assumption that the
LPA does not change much between those two definitions. While the IMD monsoon forecasts
can provide a broad regional understanding of the monsoon conditions, supplementing them with
targeted crop-specific forecasts such as ours will help improve agricultural planning and regional
water management.

We define a *strong forecast* as a forecast in which the probability assigned to one of the
two categories is at least 60%. In our situation, ten out of the thirteen years witnessed strong
forecasts. A weak forecast runs the risk of being less informative to decision-makers, whereas a
strong forecast is much more assertive and definitive, and hence decisions can be made more
easily with a strong forecast. The forecasts were also correct for seven of these ten years, as seen
in Table 2. The forecasts were correct, but barely weak, for two years (2001 and 2011). If one
considers acting only if the probability associated with a CDI forecast is at least 60%, then the
forecast is correct seven out of ten times. Raising this to 66% leads to four out of six years
classified correctly.

It is important to point out that one should also consider the uncertainty (column five in
Table 1) when evaluating the power of the forecasts. Knowing the uncertainty is useful since
years in which the uncertainty in the forecast is low and there is a strong indication for CDI may
lead to different risk management actions than years in which the forecast has strong directional
change but is also marked by high uncertainty.

### 5.2: Discussion of Results: The Utility of Targeted Forecasts

It is natural to ask how one might go about using CDI forecasts. Here is a short example
of how these forecasts can facilitate decision-making. In 2001, irrigating, or ensuring water
storage equal to 294,745 gallons per acre for the potatoes would have been the ideal situation, as
this is equivalent to being 14.4% above the average CDI value of 241 mm of water storage
equivalent. However, this exact amount cannot be known in the absence of the observed CDI
anomaly, which is found in column four of Table 1. Using the median as a plausible estimate for
the true anomaly value, roughly 268,980 gallons per acre would have been irrigated or stored
instead. A more risk-averse decision-maker may choose to use the upper quartile or even
maximum of the kNN-generated sampling distribution as a proxy for the true anomaly value.
Such decisions are often made on the basis of prior experience.

Although total seasonal rainfall is sometimes used for agricultural water planning, CDI
boasts a significant advantage over total seasonal rainfall in this capacity. CDI reliably accounts
for water stress incurred by haphazard and erratic patterns of rainfall during the season. A total
seasonal rainfall forecast that indicates a growing season with sufficient rainfall will not be
reliable when rain throughout the season is erratically distributed in clusters of rainy days,
whereby all of the rainfall in a given season occurs within a portion of the season, and the



remainder of the season is virtually dry. This is a common occurrence in monsoonal climates,
and may have deleterious effects on crops that are vulnerable to prolonged dry periods and/or
chunks of time during which rainfall is excessive. Long dry spells throughout the season that
can be detrimental to drought-sensitive crops are not accounted for in a measure of total seasonal
rainfall, making it possible for the seasonal rainfall to appear sufficient due to sporadic
occurrences of large precipitation events. Consequently, it can also serve as a better indicator
than regional rainfall to devise index insurance products for agriculture, where crop specific
indices can be developed (Skees, 2016). These characteristics of crop water stress must be
accounted for in the proper planning and management of agricultural water resources.
663   To illustrate the above point further, we appeal to Figure 5. In this figure, the varying
rainfall distribution is indicated by the vertical bars, the crop demand is given by the horizontal
line (primary y-axis), and the time series shows the cumulative deficit. The second panel shows
two distinct years during which the total seasonal rainfall was 590 mm (vertical line). During
one of these two years, the CDI value was 111 mm of water deficit for the potato crop, while the
CDI value for the other year was 228 mm. This indicates that the water stress for a particular
crop relies on both the magnitude and frequency of seasonal rainfall. When daily seasonal
rainfall is more uniform, the daily deficit values do not have the chance to accumulate as much
as when rainfall is less uniform and, as a result, when there are persistent dry spells or long
precipitation-inactive periods. Panel three shows the resulting cumulative deficit when daily
rainfall occurs with greater frequency during the JJAS season and hence the total seasonal
rainfall is distributed among the days of the growing season fairly uniformly. The fourth panel,
immediately to the right of the third panel, shows the resulting cumulative deficit when rainfall is
dominant during the first and last months of the JJAS season. While rainfall events do occur in
between, the magnitude of the rainfall is quite low, allowing the seasonal daily CDI time series
to spike to a considerably higher maximum value (228 mm) than the CDI time series in panel
three (111 mm maximum). The CDI time series recedes and recovers at the end of the season
when the rainfall increases in magnitude. Hence, CDI can discriminate between two monsoon
seasons which have the same total rainfall, but differ in that one may have rainfall distributed
uniformly over the season through modest rainfall events, while the other may have a few intense
rain events separated by long dry periods. As we can see, the latter gives rise to a much higher
CDI.

**6. Summary and Conclusion**
687   A novel crop water stress index, the CDI, was developed here as a way of estimating
water storage and irrigation requirements in the interest of agricultural water resources. As
management of water resources requires advance knowledge of water risk, the main task
accomplished here was the forecasting of CDI as an effective method for understanding and
hedging risk. This concept of forecasting CDI for evaluating irrigation requirements was applied
to a case study in the Satara district of Maharashtra, India in which the CDI pertaining to
potatoes grown in Satara during the Southwest monsoon season was forecasted using large-scale
climate indices as predictors in a semi-parametric k-nearest neighbors stochastic model that
issues probabilistic forecasts. The climate indices used were defined either concurrent to the
monsoon season or three to six months prior. Based on the hit and false alarm rates, the results
achieved using our methodology were more favorable than precipitation forecasts conducted by
the India Meteorological Department. We also observed in our method a greater tendency
towards strong and informative forecasts.





This study developed a framework for quantifying and analyzing climate-induced
agricultural risks.  It is based on (a) developing CDI for assessing crop-specific water risk,
irrigation requirements and water storage needs for the agricultural sector; (b) investigating the
sources of predictability for this indicator, and (c) developing statistically verifiable models for
issuing season-ahead probabilistic forecasts for evaluating water risk and irrigation needs.  We
can conclude that this is a useful approach to investigating irrigation requirements and that
bootstrap-based uncertainty estimation is useful for developing probability-based management
models for optimizing agricultural decisions.

**Acknowledgements**
This research was supported by:
(a) NSF grant 1360446 (Water Sustainability and Climate, Category 3)
(b) PSC-CUNY award 69729-00 47
Partial support for the third and fourth authors is provided from PepsiCo Inc. through the
WATER RISKS AND SUSTAINABILITY grant.  The statements contained within the
manuscript/research article are not the opinions of the funding agency or the U.S. government
but reflect the authors' opinions.

**Data Availability**
The CDI data used in this paper is available upon request of the contact author.

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



# Tables



**Table 1**

| Year | Probability of Above Mean | Probability of Below Mean | Observed CDI Anomaly (%) | Boxplot IQR (vertical axis units of %-anomalies) |
|---|---|---|---|---|
| 2001 | 0.59 | 0.41 | +14.4 | 10.9 |
| 2002 | 0.42 | 0.58 | +15.5 | 21.0 |
| 2003 | 0.20 | 0.80 | +37.8 | 23.1 |
| 2004 | 0.35 | 0.65 | -20.1 | 7.70 |
| 2005 | 0.25 | 0.75 | -51.3 | 12.1 |
| 2006 | 0.37 | 0.63 | -47.9 | 10.0 |
| 2007 | 0.37 | 0.63 | -20.5 | 2.60 |
| 2008 | 0.75 | 0.25 | -6.33 | 19.1 |
| 2009 | 0.64 | 0.36 | -30.0 | 5.10 |
| 2010 | 0.18 | 0.82 | -56.4 | 31.1 |
| 2011 | 0.58 | 0.42 | +2.72 | 0.19 |
| 2012 | 0.68 | 0.32 | +25.4 | 9.90 |
| 2013 | 0.18 | 0.82 | -9.36 | 24.6 |




**Table 2**

| Year | Forecast | Actual Observation | Result |
|------|----------|--------------------|--------|
| 2001 | AM (59%) | AM | Hit |
| 2002 | BM (58%) | AM | Miss |
| 2003 | BM (80%) | AM | Miss |
| 2004 | BM (65%) | BM | Hit |
| 2005 | BM (75%) | BM | Hit |
| 2006 | BM (63%) | BM | Hit |
| 2007 | BM (63%) | BM | Hit |
| 2008 | AM (75%) | BM | False Alarm |
| 2009 | AM (64%) | BM | False Alarm |
| 2010 | BM (82%) | BM | Hit |
| 2011 | AM (58%) | AM | Hit |
| 2012 | AM (68%) | AM | Hit |
| 2013 | BM (82%) | BM | Hit |




**Table 3**

| Year | CDI Forecast Results | IMD Precipitation Forecast | Actual Precipitation | IMD Forecast Results |
|---|---|---|---|---|
| 2001 | Hit | 96% of LPA | 93% of LPA | Hit |
| 2002 | Miss | Not Available | 68% of LPA | NA |
| 2003 | Miss | 99% of LPA | 40% of LPA | Miss |
| 2004 | Hit | 103% of LPA | 160% of LPA | False Alarm |
| 2005 | Hit | Not Available | 160% of LPA | NA |
| 2006 | Hit | 90% of LPA | 141% of LPA | False Alarm |
| 2007 | Hit | 96% of LPA | 163% of LPA | False Alarm |
| 2008 | False Alarm | Not Available | 95% of LPA | NA |
| 2009 | False Alarm | Not Available | 212% of LPA | NA |
| 2010 | Hit | 99% of LPA | 199% of LPA | False Alarm |
| 2011 | Hit | 98% of LPA | 85% of LPA | Miss |
| 2012 | Hit | 96% of LPA | 46% of LPA | Miss |
| 2013 | Hit | 98% of LPA | 150% of LPA | False Alarm |




960                                            **Figures**

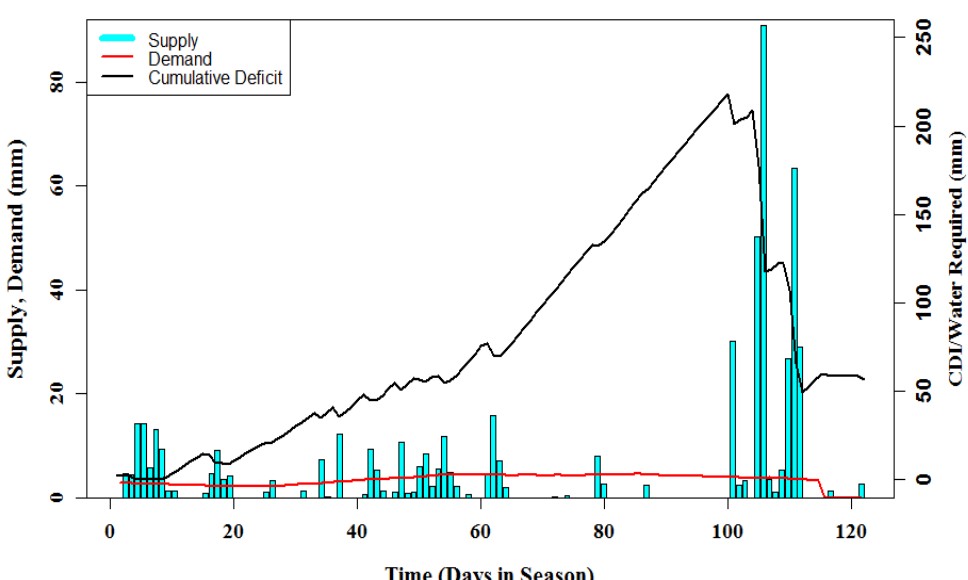

**Figure 1**















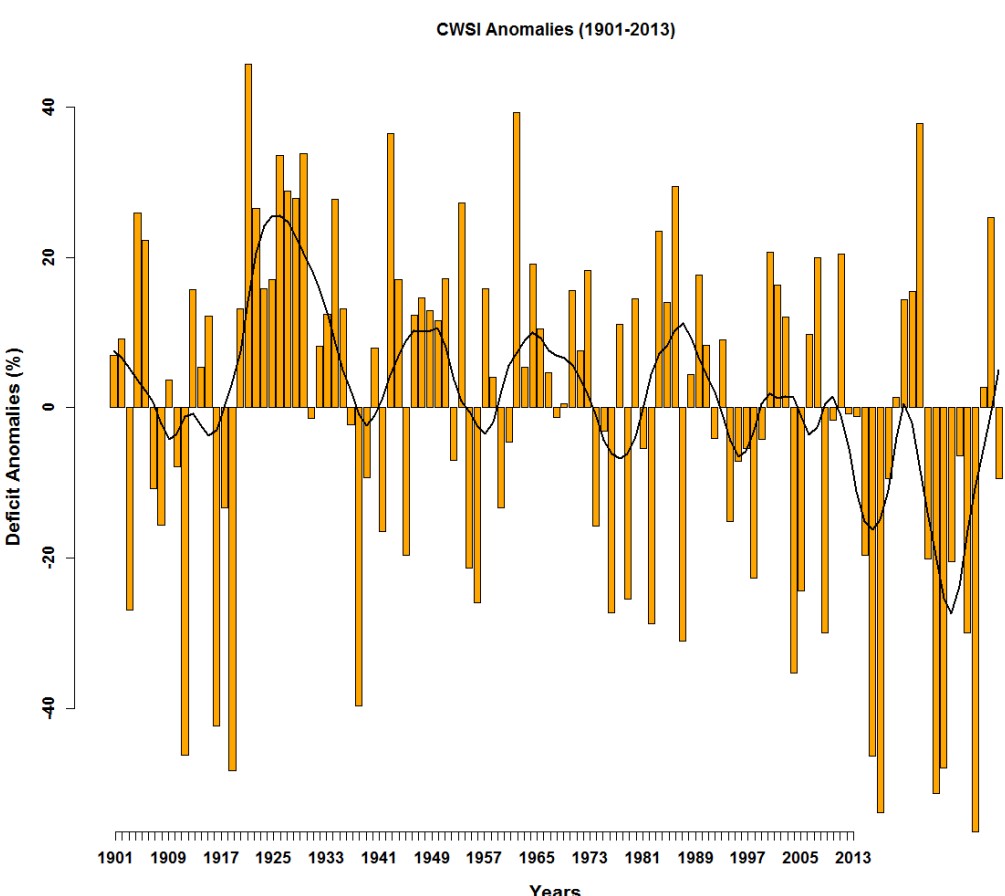


**Figure 2**












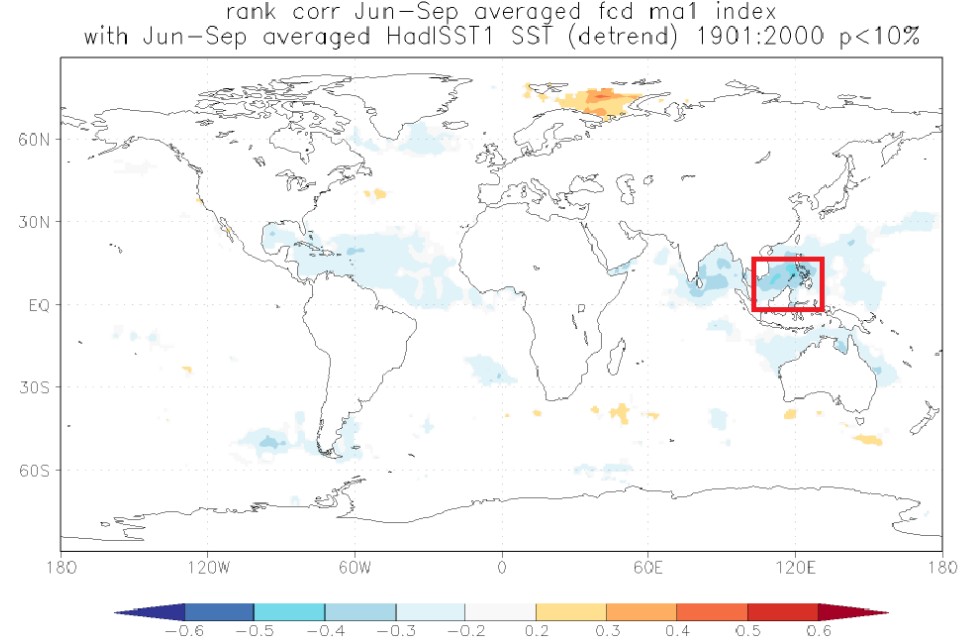


**Figure 3**

















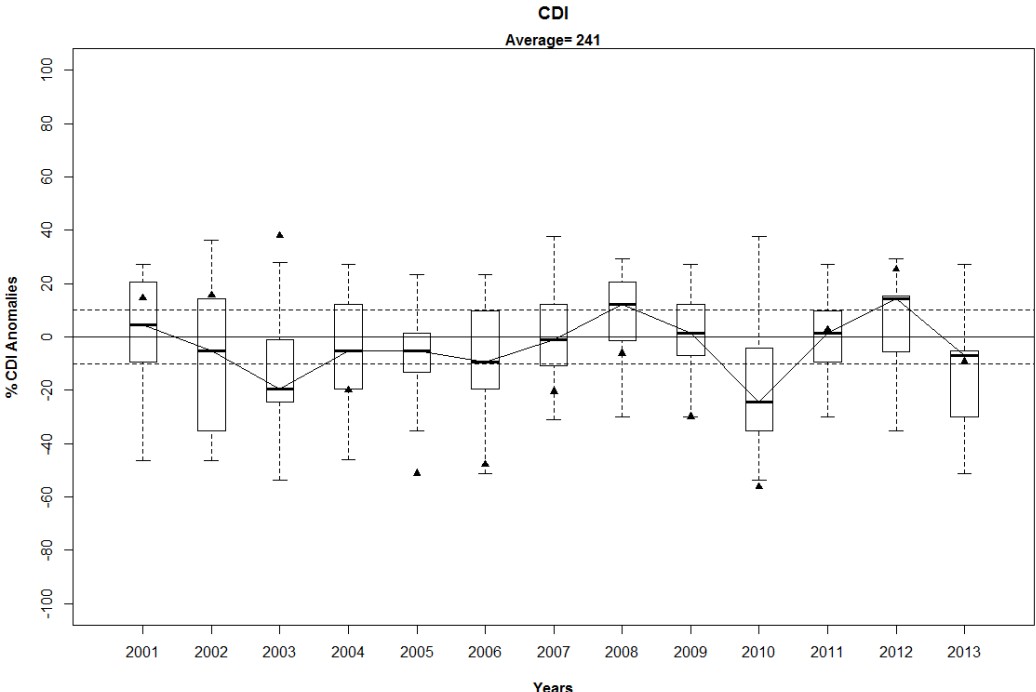

**Figure 4**














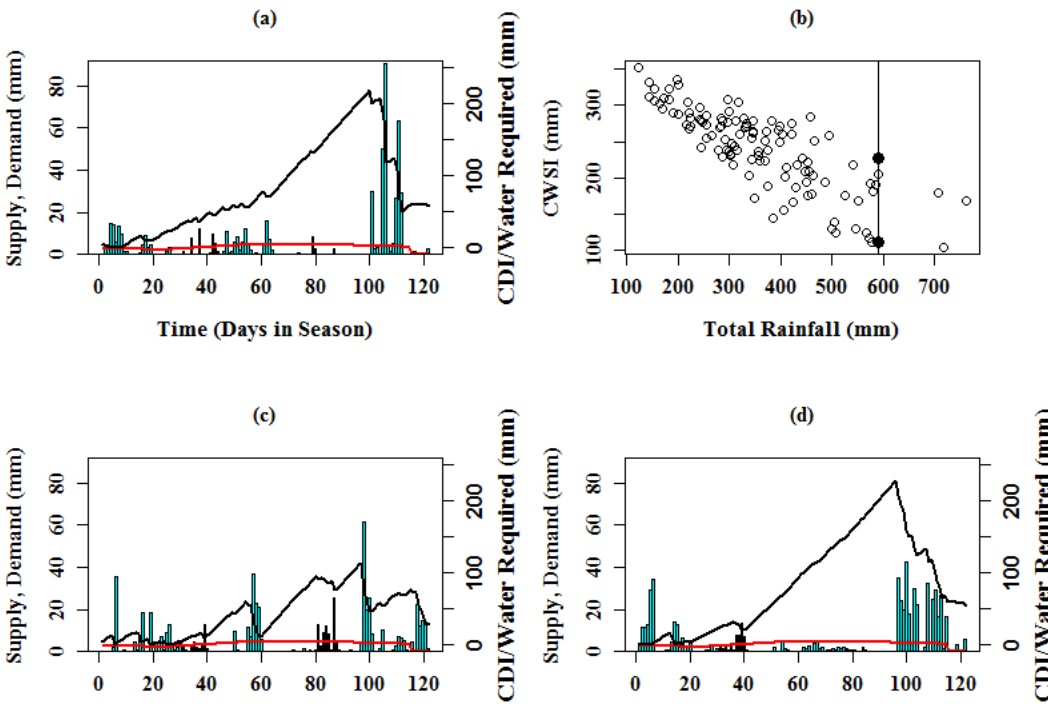

**Figure 5**













**Figure and Table Captions**

**Table 1:** The table below shows important statistics calculated from kNN forecasts of CDI.  In particular, column 2 displays the probabilities of the CDI for a particular season being above the CDI climatology.  These probabilities are calculated from the kNN sampling distribution, which in turn is simulated from historical values of the CDI based on the nearest neighbors determined in the predictor variable space.  Column 3 shows the complementary probabilities of being below this historical average.  The forecasts for years 2001-2013 are retrospective and may serve as a cross-validation for the kNN model.  Column 4 shows the values of the actual (observed) CDI anomalies with respect to the 1901-2013 climatology as percentages.  A negative value implies that the actual CDI value was below the historical average by the given percentage.  The rounded IQR values are shown in the final column of the table.

**Table 2:** The results of the kNN-generated CDI forecasts, including the most likely category (AM = Above Mean, BM = Below Mean) along with the corresponding kNN-assigned probability value expressed as a percentage in parentheses next to it (column 2), the category in which the observed anomaly value resides (column 3), and the hit/miss/false alarm designations corresponding to these results (column 4).

**Table 3:** A comparison of the CDI forecasts and the JJAS total seasonal precipitation forecasts generated by the India Meteorological Department (IMD).  Column 2 is a repeat of column 4 in Table 2; a record of the accuracy of CDI forecasts expressed in terms of hits and misses.  Column 3 contains the forecasts issued by IMD, and column 4 are the actual observations of JJAS seasonal total rainfall using rainfall data from the Satara district itself.  The fifth and final column of Table 3 shows the accuracy of the IMD forecasts in terms of hits and misses using their own 5-category system.

**Figure 1:** A plot of the cumulative deficit index (CDI) for the JJAS season in a randomly selected year in our data set.  The plot depicts the change in CDI as rainfall distribution and crop water requirement varies over the given monsoon season.  The vertical cyan bars are the daily rainfall magnitudes, the slowly-changing red line is the crop water requirement (demand) and the black time series is the CDI itself.  Notice how CDI increases as rainfall is either low in magnitude or sparsely distributed in certain blocks of time in the season.

**Figure 2:** Bar plot showing the CDI percent deficit anomalies for each of the years/growing seasons under consideration (1901 – 2013).  The black, smooth time series is produced by an 11-year LOWESS smoothing of the CDI percent deficit anomalies and is meant to show the critical trends in the CDI over the entire 1901 – 2013 period.

**Figure 3:** Spearman rank correlation between CDI in Satara and SST field during the same JJAS season.  SST region in the Indian Ocean (red box) that influences the CDI has a statistically significant correlation at the 95% significance level.

**Figure 4:** Boxplot diagrams depicting the kNN forecast distributions for CDI over the years 2001 – 2013 for potatoes grown in the Satara district, Maharashtra, India.  Longer, more stretched out boxes indicate a greater degree of variability, or uncertainty, in the forecast





distribution.  Boxes in which the median is grossly off-center indicates that the forecast
distribution is heavily skewed.  Anomalies with respect to the climatology of the predictand were
used in the boxplot calculations.  As the results are presented in terms of the percent anomalies,
the historical average is located at zero.  The triangles represent the observations as percent
anomalies about the mean.

**Figure 5:** The four panels pictured here depict the CDI in various ways.  In panels (a), (c) and
(d), the blue bars represent daily seasonal rainfall levels (in mm), the red curve represents crop
evaporative water demand ($ET_0$) and the black time series is the CDI calculated based on this
data.  Panel (a) illustrates the basic nature of CDI using the daily seasonal CDI time series from
the JJAS growing season of 2013.  Note that this time series is specifically calculated for
potatoes grown in the Satara district of Maharashtra, India during the 2013 JJAS growing season.
Panel (b) shows a scatterplot of total rainfall across all growing seasons (1901 – 2013) and CDI
across all growing seasons.  A significant negative correlation between them is apparent from
this scatterplot (Pearson correlation is -0.8, Spearman rank correlation is -0.812, Kendall rank
correlation is -0.623).  This panel demonstrates two different growing seasons, with two different
CDI values, during which the total seasonal rainfall was the same.  Panel (c) is a seasonal CDI
time series plot corresponding to the growing season with the lower CDI value on the vertical
line in panel (b).  Panel (d) is a seasonal CDI time series plot corresponding to the growing
season with the higher CDI value on the vertical line in panel (b).