# Peer review of "Season-Ahead Forecasting of Water Storage and Irrigation # Requirements"

_Hydrology and Earth System Sciences, 2018_

## Referee Comment (RC1) · Anonymous Referee #1 · 24 May 2018

General response. This is a well-written paper that develops and demonstrates a framework for season-ahead forecasts of an irrigation-relevant index, the CDI. Although there have been other studies examining season-ahead forecasting for the agricultural sector, the significance of this paper is the demonstration of the forecasting of a decision-relevant index, rather than routinely forecasted products, such as precipitation. I recommend the paper for publication, with a few minor revisions for consideration by the authors:

1. General comment for Section 4. This section is well written, but is quite dense, making it hard to follow each step. I suggest adding a flow chart detailing the main

steps (along with the associated section #), to help the reader follow along.

1.a. Related to this point, in section 4.2.1. "Predictor Selection" I was not clear if there was any consideration for having more versus less predictors, especially since these would likely be co-linear; i.e., your best model includes Niño 12 MAM-DJF, Niño 34 MAM-DJF, and ITF. It would seem like these would have similar information, though I recognize that this is a data-driven approach (i.e., is ultimately used in the feature vector in the knn, not linear regression). Was there any penalty calculated in your metrics (i.e., RMSE and RPSS) for including additional predictors?

2. General comment for section 5.1. Your evaluation of the forecasts is effective (e.g., Figure 4, and Tables 1 & 2), especially when you compare with the precip forecasts (Table 3).

2.a. Minor comment related to this point: Figure 4 & Line 562. If possible, maybe have a color coding or symbol of the triangles to indicate the directional similarity? And add a legend to that effect? Otherwise this is hard to see. At first glance, I was looking to see if the observation was captured by the IQR.

2.b. Table 3 and lines 611. Agreed that it is important to note that your forecasts are tailored to the location, which is quite resource intensive to do for every crop, and every location. I agree that this is where a framework (such as what you have put forth) is helpful, but it may be worth highlighting that there is a rich literature on opportunities and barriers to using seasonal forecasts (see next comment: 3. General comment for section 5.2).

3. General comment for section 5.2. It is useful to provide a discussion of the utility of such forecasts. Targeted forecasts, such as those presented here, can help to increase the utility of the forecasts since they intersect with actual decision contexts (e.g., irrigation needs for particular crops). It would also be worth mentioning (briefly) that there have been studies on developing useable climate information, and mention how this study fits into that bigger picture. Dilling and Lemos (2011) have a good overview

of some of the opportunities and barriers to the use of seasonal climate forecast information (but there are other studies), which might be of interest to that end. Dilling, L., & Lemos, M. C. (2011). Creating usable science: Opportunities and constraints for climate knowledge use and their implications for science policy. Global Environmental Change, 21(2), 680–689. http://doi.org/10.1016/j.gloenvcha.2010.11.006

Other Minor comments:

Line 208: Any prior studies/experience/justification for this selection?

Next time, please put the captions beneath each figure for ease of reading.

Figure 2 – What is CWSI (plot title)? Also, the x-axis does not seem to line up properly. I see the local smoother trends indicating the variability. Did you test to see if there is any monotonic trend over the time series? From the figure it seems like recent years may be pulling it down towards a negative trend (but perhaps not stat significant). Just curious, as this might be relevant to calculating the anomalies (e.g., line 539, the anomalies being estimated from the 1901-2013 mean).

---

## Referee Comment (RC2) · Anonymous Referee #2 · 31 May 2018

[referee-annotated manuscript omitted]

---

## Referee Comment (RC3) · V. Mishra (Referee) · 10 Jun 2018

The manuscript presents a framework for the season ahead forecasting for water requirements in India using cumulative deficit index. The manuscript raises an important topic, however, has a few relevant but unanswered questions. The major comments are as following:

1) Authors use the coarse resolution (1 deg) precipitation data for the period of 1901-2004 while high-resolution data can be more appropriate for this study. High-resolution and updated precipitation data are available at 0.25 deg for the 1901-2015 period (Pai et al. 2014). Moreover, air temperature data have been obtained from NCAR. Precipitation and temperature during the monsoon season have co-variability. Therefore, authors should use both precipitation and temperature data from India Meteorology Department (IMD).

2) It is not clear to me that why the forecast was done only for 2001-2013 period? To ensure the robustness of the forecast, retrospective evaluation for a long-period (at least 30-years) is required.

3) Crop water stress has been estimated using cumulative deficit index (CDI). CDI is the difference between rainfall and crop water requirement (based on PET). This raises a major concern as CDI completely ignores the role of soil moisture persistence. If I am not mistaken, this issue has to be addressed in the revised manuscript.

4) How does CDI account for pre-season soil moisture?

5) If possible, authors should test the validity of CDI using soil moisture data. Authors may use soil moisture available from global land data assimilation system (GLDAS).

6) It is not clear to me that why did authors select the potato crop and not rice and wheat? For their analysis. The approach should be evaluated for other crops and the other regions as well.

7) The estimation of daily reference crop ET estimation is based on only maximum and minimum temperature and does not include radiation and wind. Therefore, uncertainty in ET estimation should be evaluated.

Minor comments:

1) The organization of the manuscript should be improved. There are many long paragraphs that should be shortened.

2) Lines 122-129 should be removed.

3) Lines 146– correlation values should be in two digits after the decimal.

4) Lines 133-178: long paragraph

5) Line 195: CDI will mostly overestimate the demand as it does not consider soil moisture persistence?

6) Line 208: What is the basis of effective rainfall with alpha =0.7?

7) Quality of figures can be improved.

Vimal Mishra

---

## Author Comment (AC1) · 14 Jul 2018

We thank the three referees for their valuable comments. Here are our point-by-point responses. Some comments that had similar concerns were grouped together for the response. Note the following conventions: RC = referee comments, AC = author comments (replies)

Referee 1 General response. This is a well-written paper that develops and demonstrates a framework for season-ahead forecasts of an irrigation-relevant index, the CDI. Although there have been other studies examining season-ahead forecasting for the agricultural sector, the significance of this paper is the demonstration of the forecast-

ing of a decision-relevant index, rather than routinely forecasted products, such as precipitation. I recommend the paper for publication, with a few minor revisions for consideration by the authors:

Thank you.

RC1. General comment for Section 4. This section is well written, but is quite dense, making it hard to follow each step. I suggest adding a flow chart detailing the main steps (along with the associated section #), to help the reader follow along.

RC1.a. Related to this point, in section 4.2.1. "Predictor Selection" I was not clear if there was any consideration for having more versus less predictors, especially since these would likely be co-linear; i.e., your best model includes Niño 12 MAM-DJF, Niño 34 MAM-DJF, and ITF. It would seem like these would have similar information, though I recognize that this is a data-driven approach (i.e., is ultimately used in the feature vector in the knn, not linear regression). Was there any penalty calculated in your metrics (i.e., RMSE and RPSS) for including additional predictors?

AC1: We have created the flowchart suggested by the reviewer, outlining the main steps in section 4 and labelling each step with the paper section and subsection numbers in which those steps appear.

The following flowchart is added as Figure 2 in the revised manuscript (see first attachment).

AC1a: The final set of predictors we obtained from the exhaustive search method turned out to be oceanic tele-connections that are from different Âăspatial regions over the equatorial Pacific and Indian Oceans. In this regard, Nino 34 and Nino 12 trends and ITC are not correlated in space, as Nino 34 and Nino 12 are located in unconnected regions of the Pacific, and ITC is a different phenomenon best described as a pathway for warm freshwater to move from the Pacific to the Indian Ocean. However, they may be correlated in time subject to a lag. Âă

The reason that we used all three of the predictors is that we wanted to incorporated all of the spatial features. ÂăWe demonstrate concurrent time correlations across the predictors by showing a scatterplot matrix between any and every two of the three chosen predictors. ÂăThere does not appear to be any significant correlation between Nino 12 MAMMDJF and ITC. There appears to be some degree of correlation between Nino 34 MAMMDJF and ITC. ÂăAlthough Nino 34 and Nino 12 are in disconnected regions of the Pacific, they appear to have some correlation, at least over time. We did not add any penalty to the metrics (RPSS and RMSE). See second attachment for scatterplot matrix alluded to earlier.

RC2. General comment for section 5.1. Your evaluation of the forecasts is effective (e.g., Figure 4, and Tables 1 & 2), especially when you compare with the precip forecasts (Table 3).

RC2.a. Minor comment related to this point: Figure 4 & Line 562. If possible, maybe have a color coding or symbol of the triangles to indicate the directional similarity? And add a legend to that effect? Otherwise this is hard to see. At first glance, I was looking to see if the observation was captured by the IQR.

RC2.b. Table 3 and lines 611. Agreed that it is important to note that your forecasts are tailored to the location, which is quite resource intensive to do for every crop, and every location. I agree that this is where a framework (such as what you have put forth) is helpful, but it may be worth highlighting that there is a rich literature on opportunities and barriers to using seasonal forecasts (see next comment: 3. General comment for section 5.2).

AC2a: We have elected to color-code boxplots corresponding to identical directionality as gray and the boxplots corresponding to dissimilar directionality as white. ÂăA legend to that effect has been included in the plot. Thank you for this suggestion.

Figure 4 in the paper shows this change has been implemented. See third attachment.

AC2b: The literature on opportunities and barriers to using seasonal forecasts is integrated into the manuscript in the relevant section(s). The following paragraph is added as the concluding paragraph of section 5.2.

"An interesting and excellent discussion concerning the usability of such science is found in Dilling and Lemos (2011) and several papers cited therein. ÂăIn the context of that discussion, we find that our forecasting procedure combines the "science push" and "demand pull" approaches to creating scientific usability. ÂăThe impetus for crafting the CDI and, prior to that, independently developing the k-NN algorithm, was scientific. However, the decision to combine them and apply them as we have to seasonal forecasting was made with agricultural stakeholder interests in mind. ÂăAs discussed in Dilling and Lemos (2011), the problem of overcoming informal institutional barriers to availing of such seasonal forecasts, namely the idea that current methods of forecasting through weather and climate prediction centers are the only reliable methods, is one potentially faced by our methodology. ÂăIf this is the case, this is unfortunate, as we feel that our targeted forecasting system is potentially very useful to stakeholders and decision-makers in relevant sectors."

RC3. General comment for section 5.2. It is useful to provide a discussion of the utility of such forecasts. Targeted forecasts, such as those presented here, can help to increase the utility of the forecasts since they intersect with actual decision contexts (e.g., irrigation needs for particular crops). It would also be worth mentioning (briefly) that there have been studies on developing useable climate information, and mention how this study fits into that bigger picture. Dilling and Lemos (2011) have a good overview of some of the opportunities and barriers to the use of seasonal climate forecast information (but there are other studies), which might be of interest to that end. Dilling, L., & Lemos, M. C. (2011). Creating usable science: Opportunities and constraints for climate knowledge use and their implications for science policy. Global Environmental Change, 21(2), 680–689. http://doi.org/10.1016/j.gloenvcha.2010.11.006

AC3: The literature suggested here (Dilling and ÂăLemos, 2011, etc.) has been reviewed and incorporated in the revised manuscript. More discussion on the utility of these forecasts is also added. ÂăPlease see AC2 for a reference to the change in the manuscript.

Other Minor comments:

RC: Line 208: Any prior studies/experience/justification for this selection?

AC: We have used alpha = 0.7 in our previous studies of water stress in India [Devineni et al. 2013]. The selection was based on discussions with local agricultural experts from India and some corroborative tests for similar rainfall and temperature conditions in the USA [Devineni et al. 2015]. The reference was added in section 3 to the sentence declaring the value of alpha = 0.7, as shown below: In our study, we set ÂăÂă= 0.7 (Devineni et al, 2013). Here are the references Devineni, N., Perveen, S., & Lall, U. (2013). Assessing chronic and climate-induced water risk through spatially distributed cumulative deficit measures: A new picture of water sustainability in India. Water Resources Research, 49(4), 2135–2145. doi:10.1002/wrcr.20184

Devineni, N., Lall, U., Etienne, E., Shi, D., & Xi, C. (2015). America's water risk: Current demand and climate variability. Geophysical Research Letters, 1–9. doi:10.1002/2015GL063487.

RC: Next time, please put the captions beneath each figure for ease of reading. AC: We will follow this in the corrected/updated manuscript.

RC: Figure 2 – What is CWSI (plot title)? Also, the x-axis does not seem to line up properly.

AC: The CWSI plot title is a mistake – changed it to CDI. The x-axis label has also been modified to cover the entire horizontal plotting area. ÂăThe revised Figure is shown below (see fourth attachment):

RC: I see the local smoother trends indicating the variability. Did you test to see if there is any monotonic trend over the time series? From the figure it seems like recent

years may be pulling it down towards a negative trend (but perhaps not stat significant). Just curious, as this might be relevant to calculating the anomalies (e.g., line 539, the anomalies being estimated from the 1901-2013 mean).

AC: Mann-Kendall test indicates a likely monotonic decreasing trend, with a p-value of 0.013. Trend analysis typically involves carefully understanding the causes for the trend. We wish to explore these in a later work.
* * *
[Figure]

**Fig. 1.**

[Figure]

**Fig. 2.**

**Fig. 3.**

[Figure]

**Fig. 4.**

---

## Author Comment (AC2) · 14 Jul 2018

We thank the three referees for their valuable comments. Here are our point-by-point responses. Some comments that had similar concerns were grouped together for the response. ÂăNote the following convention: RC = referee comments, AC = author comments (replies).

Referee 2

The manuscript is very well written and it was a very pleasure to read it. The proposed index - the cumulative deficit index (CDI)- is novel and original and well-motivated. The

forecast model of such index has a strong contents of innovation, and furthermore appears to be very useful from a practical/applicative point of view in agriculture water management. The approach followed to construct the forecast model of CDI is novel and scientifically sound. Results have been obtained following a rigorous and clear procedure of validation. They are convincing and confirm the validity of the proposed methodology. Therefore I recommend the publication of the manuscript in the present form. I just did very few minor comments in the file attached. The authors can choose whether to take such comments into account.

Thank you for the comments. In the revised version, we are including your suggestions.

RC1: [In reference to CDI equations in section 3] I am not sure that this is the best way to define these variables Âãbecause Dt,d t is the year, but in Dj,t j is the location. I think that these formulae could be written in more clear formalism.

AC1: We have changed the indices on the defined variables (supply, demand, rainfall, crop-coefficient, etc.) to reflect day d and location j and have done away with the year index t. ÂãThe assumption is that these calculations can be made this way in any given year. The year index logically only applies when calculating the finally CDI. Here are the revised equations as they are now written in the manuscript (see supplementary pdf):

RC2: [In reference to lines 273 - 275 in first draft of manuscript] This figure could be quoted before in paragraph 3 to clear the meaning of CDI

AC2: We have put this reference to Figure 2 at the end of Section 3 as suggested by the referee and have inserted a one sentence reminder of this Figure in the paragraph where the original reference used to be.

RC3: [In reference to lines 288 - 290 in first draft of manuscript] Comment on Page 8

AC3: We thank the referee for their input, but we feel that the meaning of this sentence is clear and elect not to make any changes here. ÂãHence, the sentence remains as

"Since agriculture tends to be one of the largest consumers of water — about seventy-percent of all the world's freshwater withdrawals go towards irrigation use (USGS, 2017), and this is in addition to what is rainfed — this is an integral part of water resources management."

RC4: [In reference to the mistaken reference to MSE instead of RMSE on page 10 first draft of manuscript, line 406]

AC4: The change from MSE to RMSE has been made. ÂăHere is the new, revised sentence:

"In this manner, a single RPSS value and RMSE value were calculated for every possible combination of the predictor variables." Âă

RC5: This is a very interesting results because you start with a very large number of predictors but your procedure reduced the number of predictors just to three, limiting in such a way the dimensionality of the problem. I think that such result should be emphasized. Furthermore a synthetic explanation about the physical reasons of the dominance of the three predictors would be useful.

AC5: We will emphasize this point. As per the predictors, we have cited relevant literature for the choices in the previous manuscript and described the physical connections when we first introduced the predictors (section 4.1.2). We now added a few lines of interpretation of the trend term as appropriate. ÂăThe sentences added are:

"Their investigation showed that the Darwin pressure anomalies decrease from DJF to MAM before the occurrence of heavy monsoon rainfall and increase prior to the occurrence of deficit monsoon rainfall."

RC6: [In reference to lines 585 - 588 of the first draft manuscript submitted] Are authors sure that these definitions should not be inverted ?

AC6: We believe that it seems intuitive to define false alarm as a situation in which the outcome is not as dire or severe as the forecast implies. In this regard, the forecast

reporting a high likelihood of above normal water stress is alarming, and this sense of alarm is rendered false and assuaged by the observation reporting below average water stress. In the situation of a miss, it is reasonable to assert that the observation reports above normal water stress, but our forecast misses this important event and instead reports high likelihood of below normal water stress. ÂăAccordingly, no change has been made and we maintain the definitions we have. Hence, the definitions remain in the original sentences:

"We say that a hit has occurred if identical directionality is observed. ÂăA miss occurs if the forecast implies below average water stress, but the observation shows above average water stress. ÂăFinally, a false alarm occurs if the forecast implies above average water stress while the observation shows below average water stress." Âă

RC 7: [In reference to Figure 2 title] What is CWSI ?

AC7: The title was mistakenly written as CWSI (an old reference to this index) and has been changed to CDI accordingly. ÂăThe revised Figure (3) is now (see first figure attachment):

Please also note the supplement to this comment:
https://www.hydrol-earth-syst-sci-discuss.net/hess-2018-183/hess-2018-183-AC2-supplement.pdf

———————————————

[Figure]

**Fig. 1.**

**Supplement:**

Deficit is estimated as the difference between the seasonal crop water requirement and effective rainfall for each crop in a given location in the season. Effective rainfall is given as

$$S_{j,d} = \alpha_j * P_{j,d} \ldots (1)$$

In Eq. (1), $P_{j,d}$ is the rainfall for a day $d$ in any given year at a location $j$. $\alpha_j$ is the parameter that determines the fraction of rainfall that can be utilized by the crops for location $j$. It accounts for losses to direct runoff, evaporation and groundwater infiltration. In our study, we set $\alpha_j = 0.7$ (Devineni et al, 2013).

The water use for a given crop is estimated based on the expected growth stage and daily evapotranspiration as

$$D_{j,d} = k_{c,d}^{(j)} * ET_{0\ j,d} \ldots (2)$$

In Eq. (2), $k_{c,d}^{(j)}$ is the crop coefficient, which is the ratio of actual evapotranspiration ($ET_a$) of a given crop under non-stressed conditions to reference crop evaporation ($ET_0$). It represents crop-specific water use at various growth stages of the crop and is typically derived empirically based on local climatic conditions (Doorenbose and Pruitt, 1977). The accumulated deficit over a season is then given as

$$deficit_{j,d} = \max(deficit_{j,d-1} + D_{j,d} - S_{j,d}, 0) \text{ where } deficit_{j,d=0} = 0 \ldots (3)$$

$$CDI_{j,t} = \max(deficit_{j,d(y)} : d = 1 : n_s; t = 1 : n) ; \text{ where } deficit_{j,d(0)} = 0, \text{ y=1,...,n} \ldots (4)$$

In equation (3), $deficit_{j,d}$ refers to the accumulated daily deficit for any given year with a crop growth period of $n_s$ days in the year, $D_{j,d}$ to total daily water demand, $S_{j,d}$ to the total daily effective rainfall, for geographical location $j$, and day $d$; $t$ refers to a calendar or cropping year; and $n$ is the total number of years in the analysis. For an $n$-year record, seasonal water stress is evaluated as the maximum cumulative deficit each season and defined here as $CDI_{j,t}$.

---

## Author Comment (AC3) · 14 Jul 2018

We thank the three referees for their valuable comments. Here are our point-by-point responses. Some comments that had similar concerns were grouped together for the response. ÂăNote the following convention: RC = referee comments, AC = author comments (replies).

Referee 3

The manuscript presents a framework for the season ahead forecasting for water requirements in India using cumulative deficit index. The manuscript raises an important

topic, however, has a few relevant but unanswered questions. The major comments are as following:

RC1) Authors use the coarse resolution (1 deg) precipitation data for the period of 1901 - 2004 while high-resolution data can be more appropriate for this study. High-resolution and updated precipitation data are available at 0.25 deg for the 1901-2015 period (Pai et al. 2014). Moreover, air temperature data have been obtained from NCAR. Precipitation and temperature during the monsoon season have co-variability. Therefore, authors should use both precipitation and temperature data from India Meteorology Department (IMD).

AC1: The 0.25 degree rainfall data is not available to us at this point. We have tried to get it, but were not successful. Hence, we used the dataset we had from our previous studies. Moreover, the original citation of NCAR for temperature was done by mistake. For this work, we in fact used the temperature data from IMD. We corrected this in the revised manuscript.

RC2) It is not clear to me that why the forecast was done only for 2001-2013 period? To ensure the robustness of the forecast, retrospective evaluation for a long-period (at least 30-years) is required.

AC2: The forecast model was run over the years 1901 - 2013, but the results are only displayed for the final 13 years. ÂăThis is due to the fact that the model-forecasted ITF runs only from 2001 - 2013 and the model-forecasted Nino 34 JJAS (concurrent season Nino 34 index) runs only from 2002 - 2013. ÂăThe first 100 years (1901 - 2000) of data are used to train the model and the remaining years (2001 - 2013) are forecasted for CDI in the following way: 1901 - 2000 data for climate and CDI are used to generate a probabilistic forecast of CDI for 2001, then 1901 - 2001 data is used to generate a probabilistic forecast of CDI for 2002, and so on up until 2013. ÂăThe model-generated estimates of ITF and Nino 34 JJAS (both indices are for the concurrent JJAS season) are used for the year being forecasted, whereas observations

are used in the nearest neighbors training scheme. This is updated in this manner for each successive year. So the limitation of model-generated ITF and Nino 34 JJAS values is the limiting factor here. ÂăHowever, we have used observations for ITF and Nino 34 JJAS to generate forecasts for the years 1976 - 2000 and augmented this with the 2001 - 2013 forecasts. The table attached as supplementary PDF material summarizes the results of this longer-term evaluation. We observed 24/38 hit rate (63% hits), 9/38 false alarm rate (24% false alarms) and 5/38 miss rate (13% miss), as reported in the final row of the table (see supplementary material 1). We added this summary in the revised manuscript, in section 5.1 end. ÂăThis proves robustness of the forecasting system developed in this paper.

The sentences added to the manuscript (end of 5.1) are

"To conclude, we used observations for ITF and Nino 34 JJAS to generate CDI forecasts for the years 1976 - 2000 and augmented these forecasts with the 2001 - 2013 CDI forecasts depicted in Figure 5. ÂăRunning the forecasts for a longer period of time, which in this case is 38 years, ensures robustness of the procedure. ÂăThe hit/false alarm/miss rates resulting from this extended retrospective, adaptive forecast are 24/38 hits, 9/38 false alarms and 5/38 misses, respectively. ÂăHence, we are observing 63% hits, which indicates a fairly good, robust forecasting procedure for an informative crop water stress index."

RC3) Crop water stress has been estimated using cumulative deficit index (CDI). CDI is the difference between rainfall and crop water requirement (based on PET). This raises a major concern as CDI completely ignores the role of soil moisture persistence. If I am not mistaken, this issue has to be addressed in the revised manuscript.

AC3: CDI is a storage metric (surrogate for the amount of water that should be kept in storage in order to satisfy crop water requirement for the growing season given seasonal rainfall conditions). ÂăCDI is, in this sense, essentially a measure of soil moisture deficit. Crop water demand incorporates the temperature and crop coefficient,

which in turn represents crop water use at various stage of growth. ÂăThe precipitation is multiplied by the fraction of usable water by the crop. This multiplier implicitly takes soil moisture into account. In our still unpublished work (Troy et al. 20xx), we find comparable results with a simple bucket style water balance model that included soil moisture as a storage layer with specified porosity and allows for lag responses. To keep the index simple and with minimum amount of data, we derived this cumulative deficit index.

Troy, T, J., N Devineni and U.Lall, (20xx), A system out of balance: (Un)sustainable water resources in India's breadbasket, to be submitted to Hydrology and Earth System Sciences.

RC4) How does CDI account for pre-season soil moisture?

AC4: The CDI is based on a storage estimation algorithm (sequent peak method) which assumes that the reservoir is full at the beginning or has the potential to fill up in a few cycles. In our application, one can make a similar assumption that farmers wait for a couple of pre-monsoon rainfall events to fill up the soil store before planting, basic irrigation and tilling. In this regard, the farmer starts with reasonably full storage and the CDI will estimate the irrigation water required through the season to sustain a healthy crop.

RC5) If possible, authors should test the validity of CDI using soil moisture data. Authors may use soil moisture available from global land data assimilation system (GLDAS).

AC5: There is a statistically significant correlation between average soil moisture and CDI. ÂăGLDAS soil moisture data for surface soil moisture (SM_S), root zone soil moisture (SM_RZ) and profile soil moisture (SM_P) is obtained. ÂăThe data ran from 1948 - 2014, whereas our CDI data runs from 1901 - 2013. The GLDAS data for each of the three soil moisture types was on the daily resolution. ÂăWe calculated JJAS seasonal averages for each of SM_S, SM_RZ AND SM_P for the years 1948 - 2014, and then

correlated these three average time series with CDI. The correlation values were -0.31, -0.33 and -0.28 for the correlation of CDI with SM_S, SM_RZ and SM_P, respectively. ÂăThese are statistically significant, and indicate that CDI is a sufficiently good measure for crop water deficit and takes into account some degree of soil moisture deficit (note the negative sign of the correlation values). We also calculated JJAS seasonal maxima (i.e. for each JJAS season in the years 1948 - 2014, we took the maximum value of soil moisture) and correlated these maxima with CDI. ÂăFor SM_S, its correlation with CDI is -0.42; for SM_RZ, its correlation with CDI is -0.45; and for SM_P, its correlation with CDI is -0.37. These indicate a strong relationship between soil moisture, particularly root zone which is the most appropriate comparison, and CDI. For the future, it is possible to include soil moisture (most likely root zone) as a second variable to CDI in a multivariate forecasting system.

RC6) It is not clear to me that why did authors select the potato crop and not rice and wheat? For their analysis. The approach should be evaluated for other crops and the other regions as well.

AC6: In this paper, we were focused on presenting the forecasting system, which includes a non-linear, non-parametric algorithm (kNN) that gives a measure of uncertainty in its forecasts and a hydrologic index that fairly assesses crop water stress. ÂăThe purpose of the paper was not to extend this system to more crops and locations, which may be beyond the scope of what is required to demonstrate the procedure. However, extending this analysis to other regions and crops is an excellent idea, and we would be very interested in working with the referee on such a project. We did look at some predictors that could be used for predicting deficit in rice. The map below shows the field correlations with global SSTa. The identified predictors for rice are similar to the ones identified for Potatoes. Hence, we can use these predictors to forecast rice deficit in the same spirit. ÂăThis is shown in the attached graph (first attachment).

RC7) The estimation of daily reference crop ET estimation is based on only maximum and minimum temperature and does not include radiation and wind. Therefore, uncertainty in ET estimation should be evaluated.

AC7: We agree that we had a constrained method. ÂăTemperature and precipitation data were all that were available, so we constructed a form of CDI that fairly evaluates crop water stress yet does so with minimal, easy-to-obtain inputs. ÂăPlease see Etienne et al. (2016) for a form of CDI that incorporates Penman-Monteith instead of Hargreaves method for estimating crop water requirement.

Etienne, E., Devineni, N., Khanbilvardi, R., Lall, U., (2016). Development of a Demand Sensitive Drought Index and its Application for Agriculture over the Conterminous United States. Journal of Hydrology, 534, 219–229. doi: http://dx.doi.org/10.1016/j.jhydrol.2015.12.060

Minor comments

RC1) The organization of the manuscript should be improved. There are many long paragraphs that should be shortened.

AC1: We shortened or separated some of the longer paragraphs.

RC2) Lines 122-129 should be removed.

AC2: We prefer to keep lines 122 - 129 (of original submission); this is in keeping with HESS style and serves as a standard roadmap of the paper that people put at the end of introductions. ÂăHence, we will keep the following sentences at the end of the introduction to the paper:

"In section 2, we present a survey of the existing forecasting systems in monsoonal climates and their skill and limitations. In section 3, we discuss the background and scientific basis of CDI, including its explicit formulation and governing equations. ÂăIn section 4, we get into a thorough description of the case study and all steps involved, including background information relating to the case study and location, data collection and processing, a complete description of the forecasting model, methods and predictor selection scheme. ÂăSection 5 presents the results of the forecast, a discussion of these results and their implications, and a comparison of our results with those of IMD. Finally, section 6 summarizes and concludes the paper."

RC3) Lines 146– correlation values should be in two digits after the decimal.

AC3: We have rounded the correlation values on line 146 to two significant figures as suggested by the referee. ÂăThis was found in line 146 of the original submitted manuscript, and is now revised to the following:

"Highest skill, as measured by the correlation coefficient between observed and model-predicted SPI series, occurred at shorter lead times, with correlation values between 0.80 and Âă0.93 depending on which SPI series was forecasted."

RC4) Lines 133-178: long paragraph

AC4: We tried to shorten this.

RC5) Line 195: CDI will mostly overestimate the demand as it does not consider soil moisture persistence?

AC5: We may have overestimation of demand in a strict crop-water modeling sense, however, the partitioning of rainfall has an implicit soil moisture accounting.

RC6) Line 208: What is the basis of effective rainfall with alpha =0.7?

AC 6: We have used alpha = 0.7 in our previous studies of water stress in India [Devineni et al. 2013]. The selection was based on discussions with local agricultural experts from India and some corroborative tests for similar rainfall and temperature conditions in the USA [Devineni et al. 2015]. The reference was added in section 3 to the sentence declaring the value of alpha = 0.7, as shown below:

"In our study, we set ÂăÂă= 0.7 (Devineni et al, 2013)."

Devineni, N., Perveen, S., & Lall, U. (2013). Assessing chronic and climate-induced water risk through spatially distributed cumulative deficit measures: A new picture

of water sustainability in India. Water Resources Research, 49(4), 2135–2145. doi:10.1002/wrcr.20184

Devineni, N., Lall, U., Etienne, E., Shi, D., & Xi, C. (2015). America's water risk: Current demand and climate variability. Geophysical Research Letters, 1–9. doi:10.1002/2015GL063487.

RC7) Quality of figures can be improved.

AC7: We tried our best to have better quality.

Please also note the supplement to this comment: https://www.hydrol-earth-syst-sci-discuss.net/hess-2018-183/hess-2018-183-AC3-supplement.pdf

[Figure]

**Fig. 1.**

**Supplement:**

| Year | Results of CDI Forecast |
|------|------------------------|
| 1976 | Hit |
| 1977 | False Alarm |
| 1978 | Miss |
| 1979 | Hit |
| 1980 | False Alarm |
| 1981 | False Alarm |
| 1982 | False Alarm |
| 1983 | Hit |
| 1984 | Miss |
| 1985 | Hit |
| 1986 | Hit |
| 1987 | Hit |
| 1988 | Hit |
| 1989 | Hit |
| 1990 | Miss |
| 1991 | Hit |

| | |
|---|---|
| 1992 | Hit |
| 1993 | Hit |
| 1994 | False Alarm |
| 1995 | Hit |
| 1996 | False Alarm |
| 1997 | Hit |
| 1998 | Hit |
| 1999 | False Alarm |
| 2000 | Hit |
| 2001 | Hit |
| 2002 | Miss |
| 2003 | Miss |
| 2004 | Hit |
| 2005 | Hit |
| 2006 | Hit |
| 2007 | Hit |
| 2008 | False Alarm |
| 2009 | False Alarm |

| | |
|---|---|
| 2010 | Hit |
| 2011 | Hit |
| 2012 | Hit |
| 2013 | Hit |
| Final Statistics | 24/38 hit rate, 9/38 false alarm rate, 5/38 miss rate |